# Gestational Iron Supplementation Improves Fetal Outcomes in a Rat Model of Prenatal Alcohol Exposure

**DOI:** 10.3390/nu14081653

**Published:** 2022-04-15

**Authors:** Kaylee K. Helfrich, Nipun Saini, Sze Ting Cecilia Kwan, Olivia C. Rivera, Rachel Hodges, Susan M. Smith

**Affiliations:** 1UNC Nutrition Research Institute, University of North Carolina at Chapel Hill, Kannapolis, NC 28081, USA; kaylee.helfrich@gmail.com (K.K.H.); sk2563@cornell.edu (S.T.C.K.); olivia_rivera@unc.edu (O.C.R.); hodgesra15@ecualumni.ecu.edu (R.H.); 2Department of Nutrition, University of North Carolina at Chapel Hill, Kannapolis, NC 28081, USA

**Keywords:** prenatal alcohol exposure, iron deficiency, fetal anemia, iron supplementation, fer-in-sol, fetal growth, fetal brain development, rat iron requirement, feed conversion

## Abstract

Prenatal alcohol exposure causes neurodevelopmental disability and is associated with a functional iron deficiency in the fetus and neonate, even when the mother consumes an apparently iron-adequate diet. Here, we test whether gestational administration of the clinically relevant iron supplement Fer-In-Sol mitigates alcohol’s adverse impacts upon the fetus. Pregnant Long-Evans rats consumed an iron-adequate diet and received 5 g/kg alcohol by gavage for 7 days in late pregnancy. Concurrently, some mothers received 6 mg/kg oral iron. We measured maternal and fetal weights, hematology, tissue iron content, and oxidative damage on gestational day 20.5. Alcohol caused fetal anemia, decreased fetal body and brain weight, increased hepatic iron content, and modestly elevated hepatic malondialdehyde (*p*’s < 0.05). Supplemental iron normalized this brain weight reduction in alcohol-exposed males (*p* = 0.154) but not female littermates (*p* = 0.031). Iron also reversed the alcohol-induced fetal anemia and normalized both red blood cell numbers and hematocrit (*p*’s < 0.05). Iron had minimal adverse effects on the mother or fetus. These data show that gestational iron supplementation improves select fetal outcomes in prenatal alcohol exposure (PAE) including brain weight and hematology, suggesting that this may be a clinically feasible approach to improve prenatal iron status and fetal outcomes in alcohol-exposed pregnancies.

## 1. Introduction

Prenatal alcohol exposure (PAE) causes physical, cognitive, and behavioral deficits in affected offspring [1]. In the United States, 11.5% of pregnant women report alcohol consumption within the past month, while 3.9% report binge drinking [2]. Despite more than forty years of interventions to inform people of the dangers of alcohol consumption during pregnancy, PAE remains a significant and expensive public health issue [3,4]. Prenatal and postnatal interventions to improve PAE outcomes are thus of high priority. One potential interventional target is maternal nutrition, including maternal intake of micronutrients [5].

Iron is an essential nutrient, and during pregnancy it supports oxygen transport, energy generation, organ formation, and brain development [6]. Iron deficiency (ID) is common during pregnancy, with up to 60% of pregnant women in the U.S. becoming ID at some point during their pregnancy [7]. Clinical studies demonstrate that iron deficiency anemia (IDA) in infants exacerbates the effects of PAE on growth and that the worst growth deficits caused by PAE are found in infants with IDA [8,9]. Preclinical research in our lab has extended this work, finding that combined prenatal ID and PAE synergistically impair white matter formation and associative learning [10,11]. Additional clinical and preclinical research suggests that PAE alters iron trafficking, routing more iron for storage and less for use, ultimately causing a functional ID in the fetus [12,13]. This altered iron trafficking is likely due, in part, to alcohol’s induction of the peptide hormone hepcidin in both mother and fetus, which acts to reduce iron absorption, sequester iron, and reduce placental iron transfer [12,13,14].

Provision of additional iron during pregnancy, through dietary fortification, overcomes the functional ID caused by PAE, reversing the fetal anemia, improving brain iron, and normalizing hepcidin expression in a preclinical PAE model [15]. A limitation of that study was that the extra iron was provided prior to and throughout the pregnancy, and this does not model the more common clinical scenario wherein the ID is not addressed until the first prenatal visit or later in pregnancy [16]. Moreover, although dietary fortification is a highly effective strategy to improve iron status, provision of a perinatal iron supplement is a more clinically relevant approach.

Iron supplementation is a practical clinical intervention during pregnancy because it is effective, inexpensive, and relatively safe. Iron supplementation improves key markers of maternal iron status including serum ferritin and hemoglobin, with ID women showing the greatest improvement [17,18]. Iron supplementation is also inexpensive, costing USD 0.0086 per dose, making it practical for use in low-income populations [17,19,20]. Iron supplementation is relatively safe during pregnancy and is not associated with adverse outcomes such as maternal infection, maternal mortality, premature birth, small for gestational age, infant size, or neonatal death [21,22,23]. However, their use has limitations including mild gastrointestinal distress, which can reduce compliance [23], and a risk for iron overload for individuals with already elevated iron stores [21]. The World Health Organization recommends daily oral iron supplementation during pregnancy due to its efficacy, ease of administration, and low cost [24,25]. Here, we provide the clinical iron supplement Fer-In-Sol to alcohol-exposed pregnant rats. We hypothesized that maternal iron supplementation would normalize fetal iron status in alcohol-exposed pregnancies and have minimal adverse effects on the mother and fetus.

## 2. Materials and Methods

### 2.1. Animals and Diet

The Institutional Animal Care and Use Committee at the David H. Murdoch Research Institute approved all animal experiments, conducted in accordance with the Guide for the Care and Use of Laboratory Animals. We used an established rat model of alcohol exposure [12,15,26,27]. Nulliparous, 7-week-old female Long-Evans rats (Envigo, Indianapolis, IN, USA) were pair-housed and fed a purified, iron-sufficient diet based on AIN-76A (TD.130338, Envigo-Teklad, Madison, WI, USA; 100 mg iron per kg diet; Appendix A) and validated for iron studies in rodents [28,29]. Rats consumed this diet for a minimum of two weeks prior to breeding to standardize their nutrient content, and they remained on this diet throughout the study. Rats were time-mated at 9–11 weeks old, with the morning a plug was discovered designated as GD0.5, and were housed singly thereafter. We measured food intake and body weight on gestational days (GD) 0.5, 3.5, 5.5, 8.5, and daily from GD11.5 to GD20.5. On GD11.5, we randomly assigned dams (using the random number generator at random.org) to one of four groups: control (CON) + Water, alcohol (ALC) + Water, CON + Iron, or ALC + Iron, distributing rats equally across all four groups (*n* = 8–9 dams per treatment group). Hereafter, we refer to CON + Water and ALC + Water as simply CON and ALC. Figure 1 depicts the study design. For maternal and fetal outcomes, gestational weight gain is calculated as dam weight at GD20.5–GD0.5; non-pregnancy weight as dam weight on GD20.5–uterine horn weight; gestational food consumption as total food consumed between GD0.5–GD20.5 and placental efficiency as fetal body weight/placental weight. The food conversion ratio is calculated as total kilocalories of food consumed (GD0.5–GD20.5)/maternal weight gain.

### 2.2. Alcohol Exposure

Daily from GD13.5 to GD19.5, ALC and ALC + Iron dams received 5.0 g alcohol/kg body weight (200 proof ethanol, USP grade; Decon Labs, King of Prussia, PA, USA), given as a 31.7% solution and prepared daily in deionized water. CON and CON + Iron dams received a 43.8% maltodextrin (Envigo-Teklad, Madison, WI, USA) solution in deionized water, isocaloric to the alcohol dose. Both were administered by gastric gavage as 2 half-doses, 2 h apart, starting 2 h after lights-on at 7 a.m. At this alcohol dose, dams were sedated after the first dose and were visibly drunk from 0.5–4 h following the second dose. Because alcohol affects food consumption, we removed food from all groups 1 h prior to the first gavage and replaced it 2 h following the second gavage. We determined blood alcohol concentrations (BAC) using a separate group of pregnant rats on the first day of alcohol dosing (*n* = 3 per group, sampling blood at 15, 30, 60, 90, and 120 min following the second alcohol gavage as per [12]). Data were analyzed using repeated-measures ANOVA.

### 2.3. Iron Supplementation

Daily from GD12.5–GD19.5, the CON + Iron and ALC + Iron dams received 6 mg elemental iron/kg body weight (Fer-In-Sol, 75 mg ferrous sulfate/mL; Enfamil-Mead Johnson Nutrition, Chicago, IL, USA) by oral gavage; this represents a doubling of their daily iron intake. Control CON and ALC dams received an equivalent volume of water. The iron was administered 5–7 h following the second dose of alcohol or maltodextrin. Because rats consume most calories during the dark cycle, providing iron slightly before this time standardized iron absorption. We selected this iron dose because the equivalent iron dose in humans (300–600 mg/d) is at the upper end of the therapeutic dose of iron as a treatment for ID.

### 2.4. Tissue Collection

Fecal pellets deposited between GD19.5 and GD20.5 were frozen to quantify iron content. Dams were euthanized on GD20.5, 24–27 h following the last alcohol or maltodextrin dose. We collected maternal blood via cardiac puncture, and fetal blood via decapitation into EDTA-coated tubes, pooling fetal blood within a litter. Placentas and fetal organs (liver, brain, heart, and tail) were frozen for analysis. Dams were perfused with phosphate-buffered saline (PBS) prior to organ collection to quantify non-heme iron.

### 2.5. Hematology

We measured complete blood counts (CBCs) of dam and fetus using a pocH-100i hematology analyzer (Sysmex, Lincolnshire, IL, USA), according to manufacturer’s instructions. Blood smears from dam (*n* = 8–9 per group) and pooled fetal blood (*n* = 6–14 pups per litter per group) were stained with Wright-Giemsa stain according to manufacturer’s instructions (Millipore-Sigma, St. Louis, MO, USA). Because the hematology analyzer does not differentiate nucleated cell types, an experimenter blinded to the exposure group counted all nucleated red blood cells (nRBC) until a total of 100 white blood cells (WBCs) was reached. The nRBC count was expressed as #nRBCs/100 WBCs [30]. To quantify zinc protoporphyrin (ZnPP), we centrifuged maternal and fetal whole blood at 800× *g* for 4 min at 4 °C to remove plasma, then washed the cell pellet 3 times in PBS to remove interfering pigments. After resuspending the pellet in PBS, we mixed an aliquot with an excess of ProtoFluor Reagent^®^, and quantified ZnPP in triplicate using the ProtoFluor Z instrument according to the manufacturer’s instructions (Helena Laboratories, Beaumont, TX, USA).

### 2.6. Fetal Sex Genotyping

We determined fetal sex as described in [31], except we used rat-specific sexing primers described by [32]. We visualized PCR bands (one 250 bp band for males, two bands at 275 bp and 725 bp for females) via Azure Biosystems (Neobits Inc., Santa Clara, CA, USA).

### 2.7. Mineral Analysis

Mineral content of fetal liver, fetal brain, maternal liver, and maternal feces was quantified using inductively coupled plasma atomic emission spectroscopy (ICPOES; Children’s Hospital Oakland Research Institute, Oakland, CA, USA), sampling from each litter the female and male fetus closest to the middle of the uterine horn. Because the fetus was not perfused, we quantified the nonheme iron content of fetal liver and brain using a modified protocol from [33] using a ferrozine concentration of 1.016 mM. For all these analyses, we used *n* = 2–3 technical replicates for 8–9 litters per treatment group.

### 2.8. qPCR for Hepcidin

We sampled one dam, and one male and one female fetus per litter (*n* = 3 technical replicates for 8–9 litters per treatment group), holding uterine position constant. Methods followed the Minimum Information for Publication of Quantitative Real-Time Experiments guidelines [34]. We performed RNA isolation, cDNA synthesis, primer design, and qPCR as previously described in [26], using primer sequences from [12]. We calculated relative expression using the 2^−ΔΔCT^ method [35].

### 2.9. Thiobarbituric Acid Reactive Substances Assay

We used the thiobarbituric acid reactive substances (TBARS) assay to measure tissue malondialdehyde (MDA), testing the first two fetal livers collected from each litter (*n* = 2–3 technical replicates for 8–9 litters per treatment group). Approximately 50 mg of liver tissue was homogenized in ten volumes of homogenization buffer (50 mM TrisHCl, 150 mM NaCl, 1% NP-40, 0.5% sodium deoxycholate, 0.1% SDS, 2 μM leupeptin, 2 μM pepstatin, 10 KIU/mL aprotinin, and 400 μM PMSF). Following centrifugation at 1600× *g* for 10 min at 4 °C, samples were adjusted to the same protein content as determined using the bicinchoninic acid assay (BCA), and assayed using a TBARS assay kit (#700870, Cayman Chemical, Ann Arbor, MI, USA) according to manufacturer’s instructions.

### 2.10. Statistical Analysis

Litters having a mean size more than two standard deviations from the overall mean were excluded from analysis. This affected one CON and one CON + Iron litter, as each had two fetuses. Data were checked for normality (Shapiro–Wilk Test) and equal variance (Levene’s test). If the data did not meet one of these assumptions, we logarithmically transformed the data and proceeded with analysis. For the dam and pooled litter data, we used a two-way ANOVA, followed by Tukey’s post hoc test when the ANOVA was significant. For data from individual fetuses, we analyzed males and females separately using a mixed linear model with alcohol exposure, iron supplementation, and their interaction as independent fixed effect factors, and litter as an independent random effect factor. For all analyses, we included litter size in the models as a covariate if it interacted significantly (*p* < 0.05) with a fixed effect factor. If the mixed linear model found overall significant effects, we used Tukey’s post hoc test to determine significant differences between individual groups. For data that still could not meet the assumptions after logarithmic transformation, we performed the Kruskal–Wallis Rank Sum Test, followed by nonparametric comparisons using the Steel–Dwass method (the nonparametric version of Tukey’s test), if significant. We conducted all analyses using JMP, version 14 (JMP Inc., Cary, NC, USA) and generated all graphs in RStudio running R version 3.6.0. Outcomes were considered statistically different at *p* < 0.05. Maternal data analysis including age at breeding, gestational weight gain, weight gain from GD12.5–20.5, food conversion ratio and mean platelet volume (MPV), did not meet the assumptions of normality and equal variance. Therefore, they were analyzed by Kruskal–Wallis Rank Sum test and the Steel–Dwass post hoc test. All other maternal analyses used two-way ANOVA and Tukey’s post hoc test. For fetal data, litter size, percent survival, percent male, mean corpuscular hemoglobin (MCH), and mean corpuscular hemoglobin concentration (MCHC) data were not normal and had unequal variance. Thus, they were analyzed by Kruskal–Wallis Rank Sum test and the Steel–Dwass post hoc test. All other fetal data were analyzed by mixed linear model followed by Tukey’s post hoc test.

## 3. Results

### 3.1. Maternal Gestational Characteristics

Maternal age and weight on GD0.5 did not differ across groups (Table 1). Alcohol reduced maternal weight on GD20.5 (*p* < 0.001) and decreased gestational weight gain (CON vs. ALC, *p* = 0.032). Alcohol also reduced gestational food consumption (*p* = 0.003) and decreased the efficiency of energy conversion (*p* = 0.001), wherein ALC dams required an additional 5.1 kcal from food to gain one gram of body weight compared to CON (*p* = 0.002). Parallel calculations that included calories from the gavage generated similar findings (*p* = 0.002; data not shown). Reductions in both uterine horn weight (*p* = 0.024) and non-uterine maternal weight (*p* < 0.001) contributed to the lower weight gain in ALC dams. Iron supplementation did not affect maternal outcomes in either CON or ALC dams. Maternal BACs were 164 ± 47 mg/dL in ALC dams and 105 ± 40 mg/dL in ALC + Iron dams at 30 min after the second alcohol dose, and these values did not significantly differ (Figure 2). There was a significant effect of time (F(4,16) = 3.344, *p* = 0.036) but not group (*p* = 0.608), but was underpowered to detect differences.

### 3.2. Litter and Fetal Parameters

Alcohol did not affect litter size, percent survival, or percent male (Table 1), and it reduced uterine horn weight per fetus (*p* < 0.001). PAE reduced fetal weight in both male (−15%, *p* < 0.001; Table 2) and female (−17%, *p* < 0.001; Table 3) fetuses. In both sexes, PAE reduced absolute weights of the brain (males: −6%, *p* = 0.004; females: −4%, *p* = 0.044), liver (males: −18%, *p* < 0.001; females: −21%, *p* < 0.001), and heart (males: −10%, *p* = 0.006; females: −10%, *p* = 0.019). PAE reduced absolute placental weight only in the males (−10%, *p* = 0.012) and reduced placental efficiency in both sexes (males: −8%, *p* = 0.043; females: −11%, *p* = 0.006).

Under prenatal stress, critical organs such as the brain and heart are prioritized to receive more blood flow and nutrients, supporting their growth over less critical tissues such as liver and somatic mass [36,37]. PAE increased relative weight of the brain (males: +13%, *p* < 0.001; females: +16%, *p* < 0.001) and heart (males: +6%, *p* = 0.002; females: +9%, *p* = 0.003) compared to total body weight (Table 2 and Table 3). PAE reduced relative liver weight, but only in females (−5%, *p* = 0.039), and increased brain sparing relative to the liver in both sexes (males: +18%, *p* < 0.001; females: +24%, *p* < 0.001).

Iron supplementation did not affect most litter or fetal parameters (Table 1, Table 2 and Table 3). However, iron interacted with PAE to normalize absolute brain weight in ALC + Iron males, such that brain weight did not differ from that of controls (*p* = 0.154). In females, this benefit was less pronounced, with a trend for iron supplementation to improve brain weight in ALC females (*p* = 0.062), although the ALC + Iron brains were still smaller than controls (−6%, *p* = 0.031).

### 3.3. Maternal and Fetal Hematology

We previously reported that alcohol exposure does not affect maternal hematology in this model [12,15], and as previously, alcohol did not affect maternal RBC count (*p* = 0.878), hemoglobin (*p* = 0.478), or hematocrit (*p* = 0.254; Table 4), but it did reduce the maternal nRBC count (−27%, *p* = 0.004). Iron supplementation did not further affect maternal hematology apart from a modest drop in the mean corpuscular hemoglobin concentration (MCHC; −1%, *p* = 0.038), and a significant decline in the nRBC count (−31%, *p* = 0.001). Alcohol and iron interacted to alter the mean corpuscular hemoglobin (MCH, *p* = 0.014) such that alcohol and iron individually decreased the MCH, while ALC + Iron normalized MCH. Alcohol and iron interacted to alter zinc protoporphyrin (ZnPP, *p* = 0.043), wherein alcohol and iron individually decreased ZnPP compared to CON dams, while ALC + Iron dams had ZnPP similar to CON dams, although no differences were significant.

In contrast, PAE caused a fetal anemia characterized by reduced RBC counts (−11%, *p* = 0.028) and hematocrit (−9%, *p* = 0.030; Table 5 and Figure 3), and increased mean platelet volume (MPV, +3%, *p* = 0.011) and nRBC counts (+87%, *p* = 0.007), consistent with prior work in this model [12,15]. Iron supplementation increased hemoglobin (+4%, *p* = 0.044) and trended to normalize the RBC count (*p* = 0.058) and hematocrit (*p* = 0.056) in the ALC fetuses, indicating that iron supplementation reversed this anemia. It also increased the RBC distribution width coefficient of variation (RDW-CV, +6%, *p* = 0.012).

### 3.4. Maternal and Fetal Iron Analysis

PAE alters fetal iron trafficking and the distribution of iron across maternal and fetal tissues [12]. For the dams, alcohol did not affect fecal iron content (*p* = 0.269; Figure 4a). However, alcohol did increase the total iron content in maternal liver, although the overall effect (*p* = 0.026; Figure 4b) was driven by the ALC + Iron dams, with ALC dams being no different from CON (*p* = 1.000). Thus, alcohol exposure alone had no impact on hepatic or fecal iron content. Iron supplementation increased the fecal iron content (*p* < 0.001), consistent with prior reports [38], and it increased maternal hepatic total iron content (*p* < 0.001). Follow-up analysis using the method of Rebouche et al. [33] confirmed that 98–100% of this hepatic iron was nonheme. Supplemental iron can compete with the absorption and utilization of other divalent metals, including copper and zinc. We found no significant impact of supplemental iron on the levels of these metals in either maternal feces (Copper: *p* = 0.248, Zinc: *p* = 0.356; Appendix A) or maternal liver (Copper: *p* = 0.084, Zinc: *p* = 0.072; Appendix A). Thus, iron supplementation increased maternal hepatic iron content without adversely affecting copper or zinc content.

PAE alters fetal iron distribution, reducing its content in the fetal brain and increasing its content in liver [12,39]. Here, PAE did not affect total or nonheme iron in male livers (Figure 5a). In female livers, PAE did not significantly alter total iron (*p* = 0.326), but it increased nonheme iron content (+35%, *p* < 0.001), suggesting that alcohol increased hepatic iron stores. Iron supplementation did not affect total or nonheme iron content in either male or female livers. In fetal brain, neither PAE nor iron supplementation alone affected total or nonheme iron (Figure 5b). In the male fetal brain, PAE and iron interacted to alter total iron (*p* = 0.041), wherein the CON + Iron brains had the highest total iron, but there were no individual significant differences among groups. Neither alcohol nor iron affected the iron content in female brains Excess iron intake during pregnancy impairs fetal copper transfer and decreases maternal zinc absorption [40,41], but we found no impact of iron supplementation on the copper or zinc content of fetal liver (Copper: *p* = 0.703; Zinc: *p* = 0.352; Appendix A) or brain (Copper: *p* = 0.491; Zinc: *p* = 0.883; Appendix A).

### 3.5. Maternal and Fetal Hepcidin

Hepcidin is the master regulator of iron status and trafficking and acts to reduce iron absorption and circulation [42]. Although alcohol suppresses hepcidin production in nonpregnant adults [43,44], in the pregnant state, alcohol stimulates maternal and fetal hepcidin expression [12,13]. Consistent with this, alcohol alone increased maternal hepcidin expression by 228%, although this was not significant (CON vs. ALC: *p* = 0.127; Figure 6a). Iron also non-significantly increased hepcidin expression by 220% (CON vs. CON + Iron: *p* = 0.153) and interacted with alcohol to further increase hepcidin expression by 521% (CON vs. ALC + Iron: *p* = 0.008).

With respect to fetal hepatic hepcidin, alcohol trended to have an effect (*p* = 0.052), and interacted with sex such that hepcidin expression was slightly higher in ALC females than in ALC males (*p* = 0.043; Figure 6b). Iron supplementation had little impact on fetal hepcidin, and alcohol and iron interacted to decrease hepcidin expression in ALC + Iron fetuses but not in ALC fetuses (*p* = 0.010). No individual comparisons reached significance [12,15]. Several differences may account for this. Fer-In-Sol doubled the daily iron intake (200%) whereas dietary IF was at 500% of intake; thus, a doubling of intake may have been insufficient to meet all iron needs under PAE. Second, the IF administration spanned all of pregnancy plus the two weeks prior, suggesting a longer period may be necessary. The dietary IF would also have greater bioavailability as it is distributed throughout meals, rather than given as a single bolus. Finally, iron metabolism is tightly regulated, and those homeostatic mechanisms may have partially attenuated the supplement’s impact. However, the supplement models a midgestational intervention with a clinically relevant iron product and dosage, and thus mirrors a more realistic intervention. This, along with the absence of substantial adverse effects, supports its potential as a PAE intervention.

### 3.6. Maternal and Fetal Malondialdehyde

Excess or unbound iron is a known prooxidant [42], and both alcohol and excessive iron intakes are associated with elevated oxidative damage, especially to fatty acids, in the liver [45,46], generating malondialdehyde (MDA) as a byproduct. Alcohol trended to elevate MDA content in maternal livers (*p* = 0.059; Figure 7a) and modestly increased MDA in fetal livers (*p* < 0.001; Figure 7b). Iron supplementation further increased MDA content in maternal (*p* = 0.012) and fetal (*p* = 0.003) livers, although this represented less than a doubling of its content.

## 4. Discussion

The most important finding from this study was that gestational iron supplementation selectively reversed the negative effects of PAE upon fetal anemia and male brain weight. These effects paralleled but were not as extensive as the benefit from dietary iron fortification (IF), although the chemical form (ferrous sulfate) was the same. The supplement normalized the fetal anemia and brain weight but not hepcidin expression or brain iron content, as shown for dietary IF.

PAE altered iron-specific outcomes including fetal anemia and fetal iron content. Decreases in fetal RBC count and hematocrit, as well as an increased nRBC count, indicate that PAE caused fetal anemia, consistent with prior studies [12,15] and suggesting that not enough iron is available for use. Under such conditions iron is prioritized for hematopoiesis at the expense of other organs; for the brain this has lasting consequences including impaired energy generation and myelination, and altered neurotransmission and epigenetics [47]. PAE also increased hepatic iron storage, confirming previous findings, and this may be selective for females and perhaps reflects sex-specific growth strategies and nutrient needs [12,27]. The presence of fetal anemia suggests these hepatic iron stores may not be available for the brain or erythron [12,15], and is consistent with a functional ID, since maternal iron status was otherwise adequate. In contrast, alcohol did not cause maternal anemia or alter maternal liver iron content, but it did induce maternal hepcidin expression, consistent with previous reports in pregnancy [12]. This differs from alcohol’s impact in nonpregnant adults, where it reduces hepcidin expression [43,44]. The basis for this differential response is unclear and may reflect differences in the BMP6 and IL6 signals that control hepcidin expression [26] or an interaction of alcohol with the as-yet unidentified pregnancy factor that suppresses hepcidin as pregnancy progresses, and thus maintain an adequate iron supply [48]. In summary, PAE negatively impacted iron-specific outcomes including fetal blood iron status and hepatic iron stores, with potentially adverse consequences for prenatal and postnatal development.

Importantly, gestational iron supplementation attenuated some of PAE’s effects. Most notably, it reversed the fetal anemia caused by PAE, improving fetal RBCs, hemoglobin, and hematocrit, and mirroring the action of dietary IF [15]. Iron supplementation also improved absolute brain weight in GD20.5 PAE males, and these improvements continued postnatally, with adolescent ALC + Iron males, but not females, having brains the same weight as controls (Helfrich et al., in preparation). These findings suggest that the iron supplement’s benefit to brain weight is persistent and sex-specific, and this may reflect the male fetuses’ greater growth [27]. However, the iron supplement did not improve other measures such as fetal weight, nRBC counts, or fetal brain iron content, and emphasize that not all PAE endpoints are modifiable by improved iron status, although they may remain sensitive to ID. Importantly, these benefits of iron supplementation were conferred at a dose (6 mg/kg) comparable to the therapeutic dose provided to iron-deficient humans (300–600 mg iron sulfate per day, or 3.9–7.8 mg/kg in a 77 kg human) [49]. Although we did not perform a dose–response, these findings suggest that similar interventions may be achievable clinically. Altogether, the positive effects of gestational maternal iron supplementation on PAE fetuses supports the potential for supplemental iron as an intervention in PAE pregnancies.

We found no overtly adverse consequences of gestational iron supplementation. Although excessive iron can disrupt the absorption and metabolism of divalent metals [40,41], we found no impact upon copper or zinc content in either dam or fetus, suggesting that our iron dose was not excessive. Moreover, the hepatic iron stores and hematology of the iron-supplemented dams remained within standard rat values [50]. Iron has prooxidant effects [42,51], and although dietary IF does not cause oxidative damage in maternal livers [15], we found that iron supplementation slightly elevated the hepatic MDA content in dam and fetus. These increases are considerably lower than those documented for alcoholic iron-overload, liver damage, or ID [52,53] and are unlikely to be functionally meaningful. Although an ideal intervention for PAE would have no adverse effects, altering specifics of the iron supplementation regimen could improve iron tolerance, including alternate day dosing, gentler iron formulations, or intravenous iron [54,55,56,57]. We selected ferrous sulfate as the iron formulation because it is effective at improving iron status, safe during pregnancy and childhood, and cost-effective for low-income populations [17,21,22,24,58]. However, ferrous sulfate does exhibit more adverse side effects than newer iron formulations [51,57], and future research may consider these more efficacious alternatives to improve iron status.

Finally, because we quantified intake of a food with a fixed caloric density, we were able to determine alcohol’s impact on feed conversion rate (FCR). The increased FCR in ALC dams, regardless of whether alcohol calories are included in that determination, suggests that alcohol may require increased caloric intake to maintain gestational weight gain. Moreover, alcohol reduced maternal weight gain to a greater extent than it reduced food intake. Our finding may inform the consistent observations that, in studies of rat dams who consume a fixed-calorie liquid diet, the pair-fed dams gain weight at a similar rate to their calorie-matched alcohol-fed dams, but the fetal weights from pair-fed mothers are similar to those of ad lib-fed control mothers, whereas the alcohol fetuses remain significantly smaller than both, despite the pair-feeding [59]. This suggests that alcohol may reduce the efficiency of calorie usage, even when mothers consume the same amount of food or calories. This appears to be the first description of this finding, and it merits additional investigation.

Although the best prevention of FASD is the cessation of alcohol consumption during pregnancy, this is difficult in practice, and rates of gestational alcohol consumption remain high [2]. Alcohol increases prenatal iron needs beyond what is generally considered adequate, and perhaps beyond what is typically provided by an iron-adequate diet. This study shows that maternal iron supplementation during PAE improves select fetal outcomes including fetal hematology and male brain weight. Iron supplementation may be especially beneficial in populations where PAE and ID co-occur, as ID populations are at risk of the worst PAE outcomes [7,8,9,60,61,62]. Administration of micronutrient supplements to alcohol-using women is feasible and effective at improving neurodevelopmental outcomes in the offspring [63,64]. Iron supplements consumed during pregnancy may be one way to mitigate alcohol’s gestational impact and improve offspring outcomes.

## 5. Conclusions

In summary, we report here that a relatively inexpensive iron supplement mitigates the impact of PAE upon the fetus, including prevention of PAE-induced anemia and the normalization of brain weight for male offspring. The iron supplement was administered orally to the dam during the alcohol exposure period and at a dosage that was clinically relevant when extrapolated to human pregnancy. Minimal adverse effects were observed in dam or fetus. These data suggest that direct supplementation of the mother with iron may be a clinically feasible approach to improve prenatal iron status and fetal outcomes in alcohol-exposed pregnancies.

## Figures and Tables

**Figure 1 nutrients-14-01653-f001:**
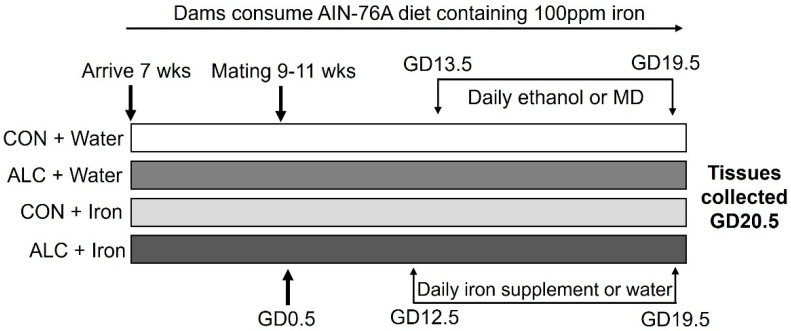
Study design. ALC, alcohol exposed; CON, control; GD, gestational day; MD, maltodextrin; wks, weeks.

**Figure 2 nutrients-14-01653-f002:**
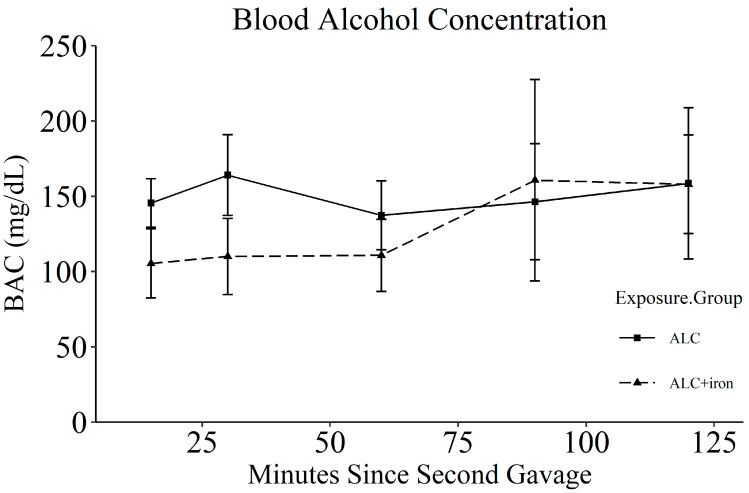
Blood alcohol concentrations of dams dosed with either alcohol (*n* = 3) or alcohol and iron (*n* = 3). Each data represents mean ± SEM. ALC, alcohol exposed; BAC, blood alcohol concentration.

**Figure 3 nutrients-14-01653-f003:**
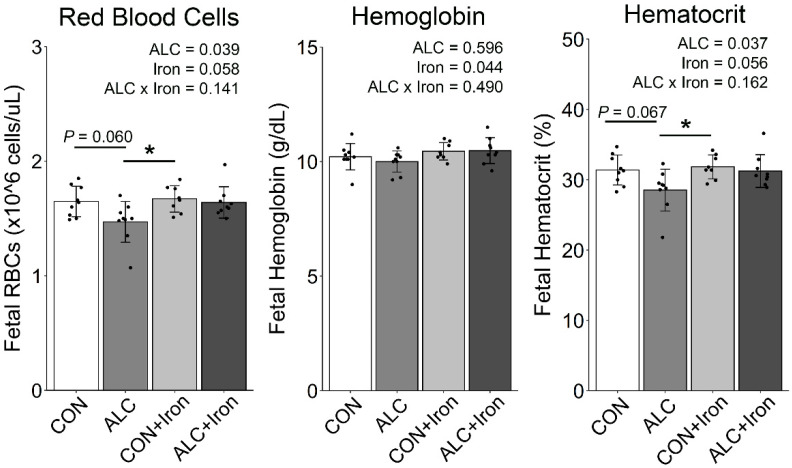
Alcohol exposure reduced fetal RBCs and hematocrit, and were normalized by iron supplementation. *n* = 8–9 litters per group (blood pooled across the litter). Each data point (black dot) represents values for individual litter, and bars show the group means ± SD. * *p* < 0.05. ALC, alcohol exposed; CON, control; RBCs, red blood cell count.

**Figure 4 nutrients-14-01653-f004:**
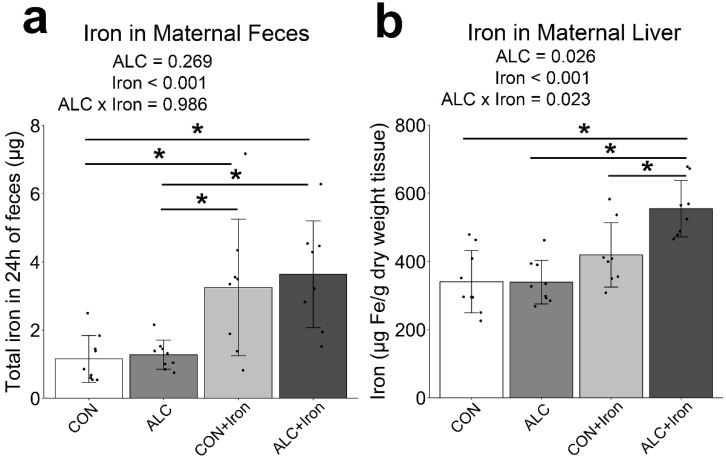
Alcohol exposure did not affect iron content in maternal feces or liver, whereas iron supplementation increased iron content in both feces and liver. Total iron in maternal feces collected over 24 h from GD19.5–20.5 (**a**), and total iron in perfused maternal liver at GD20.5 (**b**). Each data point (black dots) represents values for individual dams, and bars show the group means ± SD. * *p* < 0.05. ALC, alcohol exposed; CON, control.

**Figure 5 nutrients-14-01653-f005:**
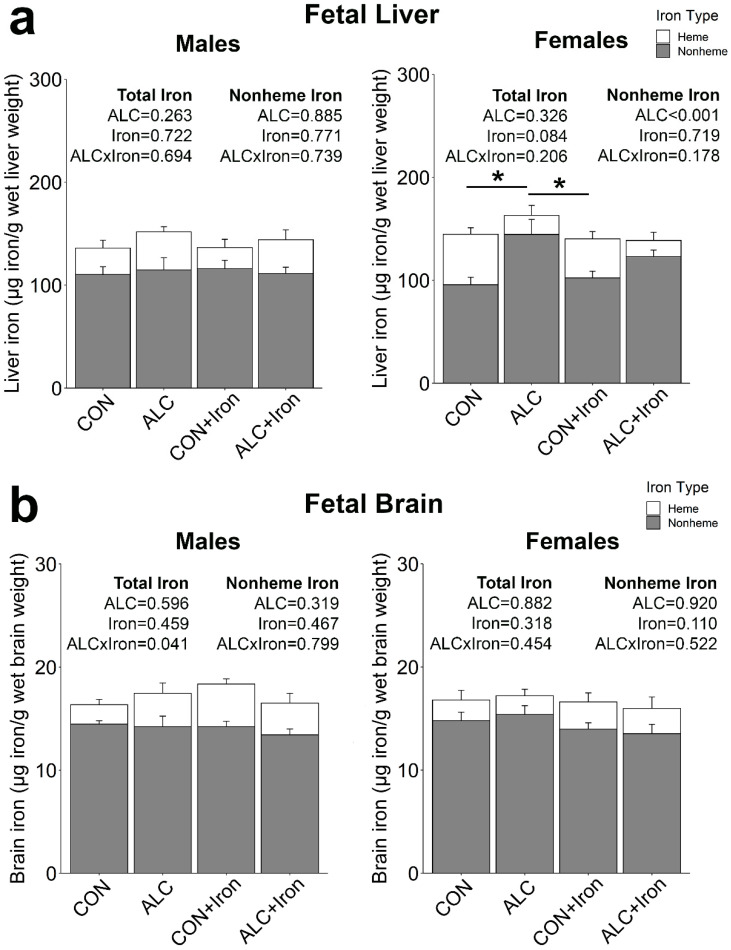
Alcohol exposure increased the iron content of female, but not male, fetal livers. Total and nonheme iron were quantified in fetal livers (**a**) and fetal brains (**b**) at gestational day 20.5. Bars represent means ± SEM. * *p* < 0.05. ALC, alcohol exposed; CON, control.

**Figure 6 nutrients-14-01653-f006:**
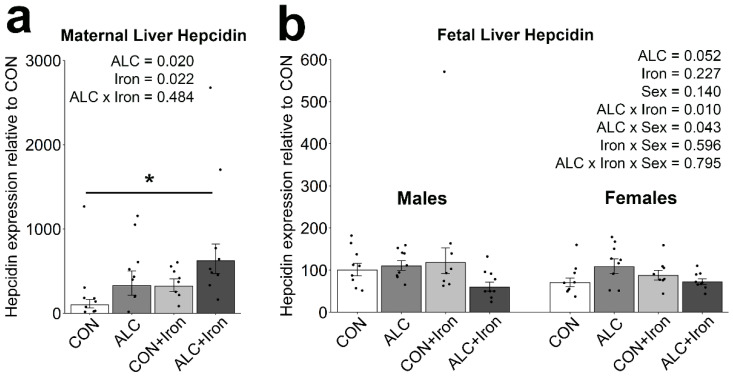
Alcohol exposure and iron induced maternal hepcidin but minimally affected fetal hepcidin. Hepcidin mRNA content was quantified in maternal (**a**) and fetal (**b**) livers at gestational day 20.5. *n* = 8–9 litters per group, sampling 1 dam, 1 male, and 1 female per litter. Individual points show the values for individual animals, and bars show the group means ± standard error. Maternal data were analyzed using two-way ANOVA and Tukey’s post hoc test, and fetal data were analyzed by mixed linear model and Tukey’s post hoc test. An asterisk denotes a significant difference at * *p* < 0.05. Abbreviations: ALC, alcohol exposed; CON, control.

**Figure 7 nutrients-14-01653-f007:**
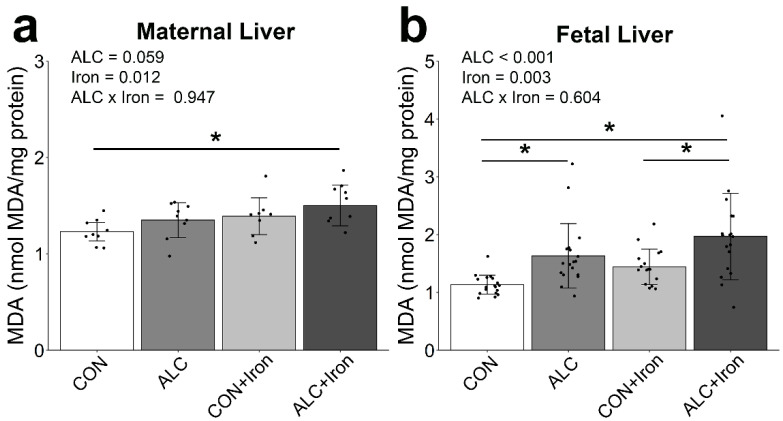
Alcohol exposure and iron supplementation modestly elevate MDA content in maternal and fetal liver. MDA content was quantified in maternal (**a**) and fetal (**b**) livers at gestational day 20.5. *n* = 8–9 litters per group, sampling 1 dam, 1 male, and 1 female per litter. Data points (black dots) show results for individual animals, while bars show group means ± standard error. For fetal liver, male and female data were combined because there was no effect of sex. Data were analyzed by two-way ANOVA and Tukey’s post hoc test. An asterisk denotes a significant difference at * *p* < 0.05. Abbreviations: ALC, alcohol exposed; CON, control; MDA, malondialdehyde.

**Table 1 nutrients-14-01653-t001:** Alcohol exposure, but not iron supplementation, altered maternal and litter outcomes.

	CON	ALC	CON + Iron	ALC + Iron	*p* Value		
					ALC	Iron	ALC × Iron
**Maternal Outcomes**							
Age at Breeding (wks)	10.8 ± 0.6	10.2 ± 0.6	10.3 ± 0.8	10.5 ± 0.9	Overall *p* = 0.195		
Weight on GD0.5 (g)	236 ± 18	226 ± 19	230 ± 20	232 ± 15	0.490	0.955	0.347
Weight on GD20.5 (g)	345 ± 18 ^a^	307 ± 20 ^c^	339 ± 29 ^ab^	313 ± 24 ^bc^	**<0.001**	0.962	0.442
Gestational weight gain (g)	109 ± 8 ^a^	81 ± 25 ^b^	109 ± 15 ^ab^	81 ± 28 ^b^	Overall *p* = **0.003**		
Weight gain GD12.5–20.5 (g)	50 ± 6 ^a^	24 ± 24 ^b^	54 ± 7 ^a^	25 ± 22 ^b^	Overall *p* < **0.001**		
Non-pregnancy weight (g)	294 ± 20 ^a^	258 ± 16 ^b^	284 ± 24 ^ab^	269 ± 18 ^ab^	<**0.001**	0.967	0.117
Gestational food consumption (g)	352 ± 22 ^a^	306 ± 31 ^b^	341 ± 35 ^ab^	320 ± 41 ^ab^	**0.003**	0.890	0.248
Food conversion ratio (kcal/g)	12.9 ± 1.0 ^a^	16.42 ± 4.1 ^b^	12.6 ± 0.8 ^a^	18.6 ± 10.1 ^b^	Overall *p* = **0.001**		
**Litter Outcomes**							
Litter Size (*n*)	10.0 ± 1.8	11.2 ± 1.1	10.9 ± 1.7	9.8 ± 2.8	Overall *p* = 0.477		
Percent Survival (%)	96 ± 9	93 ± 8	96 ± 5	87 ± 12	Overall *p* = 0.332		
Percent Male (%)	54 ± 20	47 ± 14	60 ± 7	45 ± 20	Overall *p* = 0.323		
Uterine Horn Weight (g)	51.0 ± 7.1 ^ab^	48.6 ± 6.9 ^ab^	55.0 ± 8.6 ^a^	43.3 ± 11.8 ^b^	**0.024**	0.830	0.127
Uterine Weight Per Fetus (g)	5.1 ± 0.4 ^a^	4.3 ± 0.4 ^b^	5.1 ± 0.3 ^a^	4.5 ± 0.5 ^b^	**<0.001**	0.767	0.426

Values are means ± SD. Means that do not share a common superscript letter differ at *p* < 0.05. Significant *p*-values are bolded. ALC, alcohol exposed; CON, control; GD, gestational day.

**Table 2 nutrients-14-01653-t002:** In male fetuses, alcohol exposure reduced absolute fetal and organ weights and increased relative weights of brain and heart, while iron improved absolute brain weight.

	MALES				*p* Value		
	CON	ALC	CON + Iron	ALC + Iron	ALC	Iron	ALC × Iron
**Absolute weight (g)**							
Fetus	3.69 ± 0.28 ^a^	3.03 ± 0.41 ^b^	3.70 ± 0.28 ^a^	3.22 ± 0.42 ^b^	**<0.001**	0.449	0.341
Placenta	0.39 ± 0.07 ^a^	0.35 ± 0.05 ^bc^	0.38 ± 0.06 ^ab^	0.34 ± 0.04 ^c^	**0.012**	0.391	0.497
Brain	0.16 ± 0.02 ^a^	0.14 ± 0.01 ^b^	0.15 ± 0.02 ^a^	0.15 ± 0.02 ^ab^	**0.004**	0.725	**0.014**
Liver	0.26 ± 0.05 ^a^	0.21 ± 0.04 ^b^	0.26 ± 0.04 ^a^	0.22 ± 0.05 ^b^	**<0.001**	0.960	0.392
Heart	0.018 ± 0.002 ^a^	0.015 ± 0.002 ^b^	0.018 ± 0.002 ^a^	0.016 ± 0.003 ^ab^	**0.006**	0.668	0.416
Placental efficiency	9.66 ± 1.43 ^a^	8.70 ± 1.66 ^b^	9.89 ± 1.21 ^a^	9.28 ± 1.22 ^ab^	**0.043**	0.232	0.921
**Relative weight (g/100 g total body weight)**							
Brain	4.3 ± 0.5 ^a^	4.8 ± 0.7 ^b^	4.1 ± 0.4 ^a^	4.7 ± 0.6 ^b^	**<0.001**	0.408	0.593
Liver	7.1 ± 1.3	6.8 ± 1.0	7.0 ± 1.0	6.9 ± 0.9	0.270	0.824	0.840
Heart	0.47 ± 0.06 ^a^	0.50 ± 0.04 ^bc^	0.48 ± 0.05 ^ab^	0.51 ± 0.05 ^c^	**0.002**	0.802	0.884
Brain to liver ratio	0.61 ± 0.11 ^a^	0.72 ± 0.17 ^b^	0.60 ± 0.11 ^a^	0.70 ± 0.14 ^b^	**<0.001**	0.833	0.969

Values are means ± SD. Means that do not share a common superscript letter differ at *p* < 0.05. Significant *p*-values are bolded. ALC, alcohol exposed; CON, control.

**Table 3 nutrients-14-01653-t003:** In female fetuses, alcohol exposure reduced absolute fetal and organ weights and increased relative weights of brain and heart.

	FEMALES				*p* Value		
	CON	ALC	CON + Iron	ALC + Iron	ALC	Iron	ALC × Iron
**Absolute weight (g)**							
Fetus	3.52 ± 0.30 ^a^	2.88 ± 0.41 ^b^	3.46 ± 0.27 ^a^	2.94 ± 0.48 ^b^	**<0.001**	0.819	0.562
Placenta	0.37 ± 0.07	0.34 ± 0.04	0.35 ± 0.05	0.32 ± 0.04	0.163	0.322	0.809
Brain	0.16 ± 0.02 ^a^	0.14 ± 0.01 ^b^	0.15 ± 0.02 ^ab^	0.15 ± 0.01 ^b^	**0.044**	0.518	0.062
Liver	0.27 ± 0.05 ^a^	0.20 ± 0.04 ^b^	0.25 ± 0.03 ^a^	0.21 ± 0.04 ^b^	**<0.001**	0.588	0.293
Heart	0.016 ± 0.003 ^a^	0.015 ± 0.002 ^c^	0.016 ± 0.002 ^ab^	0.015 ± 0.003 ^bc^	**0.019**	0.624	0.607
Placental efficiency	9.73 ± 1.51 ^ab^	8.69 ± 1.25 ^c^	10.11 ± 1.21 ^a^	9.04 ± 1.35 ^bc^	**0.006**	0.327	0.894
**Relative weight (g/100 g total body weight)**							
Brain	4.5 ± 0.4 ^a^	5.1 ± 0.7 ^b^	4.3 ± 0.5 ^a^	5.1 ± 0.7 ^b^	**<0.001**	0.619	0.654
Liver	7.6 ± 1.0 ^a^	7.1 ± 1.0 ^b^	7.2 ± 0.6 ^ab^	7.0 ± 0.8 ^b^	**0.039**	0.147	0.351
Heart	0.47 ± 0.06 ^a^	0.51 ± 0.05 ^b^	0.47 ± 0.05 ^a^	0.51 ± 0.07 ^b^	**0.003**	0.904	0.745
Brain to liver ratio	0.59 ± 0.10 ^a^	0.74 ± 0.16 ^b^	0.60 ± 0.09 ^a^	0.75 ± 0.16 ^b^	**<0.001**	0.905	0.937

Values are means ± SD. Means that do not share a common superscript letter differ at *p* < 0.05. Significant *p*-values are bolded. ALC, alcohol exposed; CON, control.

**Table 4 nutrients-14-01653-t004:** Alcohol exposure and iron supplementation minimally affected maternal hematology.

	CON	ALC	CON + Iron	ALC + Iron	*p* Value		
					ALC	Iron	ALC × Iron
RBC (×10^6^ cells/μL)	6.55 ± 0.26	6.52 ± 0.23	6.60 ± 0.19	6.61 ± 0.45	0.878	0.502	0.856
Hemoglobin (g/dL)	12.4 ± 0.5	11.9 ± 0.4	12.0 ± 0.3	12.2 ± 0.7	0.478	0.866	0.071
Hematocrit (%)	36.8 ± 1.5	35.5 ± 0.8	36.4 ± 0.7	36.6 ± 2.1	0.254	0.508	0.174
MCV (fL/cell)	56.1 ± 1.1	54.6 ± 1.2	55.2 ± 1.1	55.3 ± 1.5	0.108	0.813	0.051
MCH (pg/cell)	18.9 ± 0.5 ^a^	18.3 ± 0.4 ^ab^	18.2 ± 0.5 ^b^	18.5 ± 0.6 ^ab^	0.368	0.180	**0.014**
MCHC (pg/dL)	33.6 ± 0.4 ^a^	33.5 ± 0.4 ^a^	33.0 ± 0.6 ^a^	33.4 ± 0.5 ^a^	0.386	**0.038**	0.214
PLT (×10^3^ cells/μL)	859 ± 149	788 ± 118	881 ± 138	848 ± 130	0.265	0.375	0.677
RDW-CV (%)	13.0 ± 0.3	13.0 ± 0.4	12.9 ± 0.4	13.1 ± 0.5	0.493	0.930	0.594
MPV (fL/cell)	7.8 ± 0.4	7.7 ± 0.3	7.7 ± 0.2	7.6 ± 0.2	Overall *p* = 0.452		
ZnPP (μmol ZnPP/mol heme)	69.7 ± 5.2	65.2 ± 4.0	65.3 ± 3.6	66.7 ± 5.0	0.325	0.345	0.063
nRBC (per 100 WBCs)	204 ± 38 ^a^	157 ± 44 ^ab^	150 ± 57 ^ab^	100 ± 45 ^b^	**0.004**	**0.001**	0.902

Values are means ± SD. Means that do not share a common superscript letter differ at *p* < 0.05. Significant *p*-values are bolded. ALC, alcohol exposed; CON, control; MCH, mean corpuscular hemoglobin; MCHC, mean corpuscular hemoglobin concentration; MCV, mean corpuscular volume; MPV, mean platelet volume; nRBC, nucleated red blood cell count per 100 WBCs; PLT, platelet count; RBC, red blood cell count; RDW-CV, red blood cell distribution width coefficient of variation; WBC, white blood cell count; ZnPP, zinc protoporphyrin.

**Table 5 nutrients-14-01653-t005:** Alcohol exposure caused fetal anemia which iron supplementation resolved.

	CON	ALC	CON + Iron	ALC + Iron	*p* Value		
					ALC	Iron	ALC × Iron
RBC (×10^6^ cells/μL)	1.65 ± 0.13 ^ab^	1.47 ± 0.18 ^b^	1.67 ± 0.12 ^a^	1.64 ± 0.14 ^ab^	**0.039**	0.058	0.141
Hemoglobin (g/dL)	10.2 ± 0.6 ^a^	10.0 ± 0.5 ^a^	10.5 ± 0.4 ^a^	10.5 ± 0.6 ^a^	0.596	**0.044**	0.490
Hematocrit (%)	31.4 ± 2.1 ^ab^	28.5 ± 3.0 ^b^	31.8 ± 1.7 ^a^	31.2 ± 2.3 ^ab^	**0.037**	0.056	0.162
MCV (fL/cell)	190.6 ± 3.3	194.4 ± 4.5	190.7 ± 3.5	190.6 ± 2.8	0.138	0.138	0.123
MCH (pg/cell)	62.2 ± 4.6	69.0 ± 10.5	62.8 ± 5.3	64.1 ± 4.3	Overall *p* = 0.225		
MCHC (pg/dL)	32.6 ± 1.9	35.4 ± 4.5	32.9 ± 2.2	33.6 ± 1.9	Overall *p* = 0.273		
PLT (×10^3^ cells/μL)	135.4 ± 20.3	122.7 ± 15.7	140.4 ± 23.9	130.9 ± 23.2	0.126	0.360	0.818
RDW-CV (%)	14.9 ± 0.8 ^ab^	14.1 ± 1.1 ^b^	15.1 ± 0.7 ^ab^	15.6 ± 0.9 ^a^	0.541	**0.012**	0.052
MPV (fL/cell)	12.0 ± 0.5 ^a^	12.4 ± 0.4 ^a^	11.9 ± 0.3 ^a^	12.2 ± 0.3 ^a^	**0.011**	0.061	0.565
ZnPP (μmol ZnPP/mol heme)	65.4 ± 8.7 ^a^	59.0 ± 4.2 ^a^	59.2 ± 3.7 ^a^	61.9 ± 6.4 ^a^	0.396	0.479	**0.043**
nRBC (per 100 WBCs)	907 ± 397 ^a^	1153 ± 241 ^ab^	768 ± 393 ^a^	1985 ± 1384 ^b^	**0.001**	0.556	0.102

Values are means ± SD. *n* = 8–9 litters per group. Means that do not share a common superscript letter differ at *p* < 0.05. Significant *p*-values are bolded. ALC, alcohol exposed; CON, control; MCH, mean corpuscular hemoglobin; MCHC, mean corpuscular hemoglobin concentration; MCV, mean corpuscular volume; MPV, mean platelet volume; nRBC, nucleated red blood cell count per 100 WBCs; PLT, platelet count; RBC, red blood cell count; RDW-CV, red blood cell distribution width coefficient of variation; WBC, white blood cell count; ZnPP, zinc protoporphyrin.

## Data Availability

Not applicable.

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
