# Peer review of "Gestational Iron Supplementation Improves Fetal Outcomes in a Rat Model of Prenatal Alcohol Exposure"

_nutrients, 2022, doi:10.3390/nu14081653_

Round 1
Reviewer 1 Report
Helfich and colleagues report the effects of iron supplementation during pregnancy on the impact of prenatal ethanol exposure on maternal and fetal outcome measures.
Major comments:
Blood alcohol data analysis and interpretation: On lines 197 - 200, the authors report mean serum ethanol concentrations of 164 mg/dL (no iron) and 105 mg/dL (iron supplemented), 30 minutes after the second oral gavage dose. However, the authors collected blood samples at five different time points over the two hours after the second oral ethanol dose. The authors do not state what statistical procedure was used to make the statement that the serum ethanol levels were not different between groups and one wonders whether a repeated-measures ANOVA might have been considered to incorporate all of the data collected over time? Further, given the sample sizes were only three dams per group and the degree of variability within each group, there is concern that there may not have been sufficient statistical power to conclude that the differences were not significantly different. Examining Figure S1, one wonders whether the iron supplementation procedure, initiated the day before blood samples were collected, might have affected the gastric lining in a manner that may have limited or at least delayed the absorption of ethanol, as Figure S1 might suggest? These are critical considerations because if the means actually reflect a roughly 33% reduction in ethanol levels in the dams supplemented with iron, this presents as a potentially significant confound in the interpretation of all of the outcomes data. Finally, serum ethanol data should never be provided as supplementary data as it is a key reference measure for alcohol research.
Data presentation and statistical procedures for examining sex-related effects. How fetal data is presented by sex and statistically analyzed varies throughout the manuscript. In some cases, the data is presented in separated tables (Tables 2 and 3) or separate panels (Figure 4). All of the data where there were separate male and female samples should have been presented as shown in Figure 6B and analyzed by three-way ANOVA.
Data presentation issues: There were a number of issues that made comprehension of the data presented more challenging than need be. While the authors’ instruction for the journal suggest embedding the tables and figures within the text, there has to be a way to better distinguish data with attendant footnotes or figure legends than in the manner presented in this manuscript. This is compounded by the fact that results were stated in the context of table footnotes and in figure legends, which the authors should avoid doing. Figure legends should only provide the information necessary to define the data presented, sample sizes, statistical procedures used and define any statistical symbols present in the figure. Likewise, data tables should only contain footnotes that define the measures and their units along with definition of statistical symbols. Superscripted letters were used to footnote both measures as well as statistical differences, which was confusing. This reviewer would suggest adopting a system of symbols that is used consistently throughout all of the data tables and figures where possible. Also, the authors used standard deviation with some measures and standard deviation in other cases and did not always clearly indicate which variance measure was portrayed. One method of variance should be used throughout. Finally, the authors list p values in the data tables, which makes the tables dense, and they provide limited details on the statistical data portrayed in the figures. For the data tables, perhaps a compromise would be to parenthetically state the p values in the text of the results section only for the effects or interactions that are significant?
Minor Comments / Needed Clarifications:
Lines 64-66: Clarify how the issue of different iron formulations cause GI distress relates to the current study, as different diet formulations were not tested here.
Lines 81 – 83: On what basis was it determined that two weeks of consuming the Envigo 76A diet is sufficient to “standardized nutrient content”?
Lines 92 – 93: Generally, 200 proof (100%) ethanol is avoided in alcohol research studies given the presence of variable trace amounts of benzenes as part of the distillation process, which creates a potential confound easily avoided by using 190 proof (95%) ethanol.
Lines 100 -102: BAC measurements were made in a separate sets of rat dams (six per group) on the first day of ethanol administration (and one day after the start of iron supplementation.
Lines 170 – 174: Which data sets did not meet normality or equal variance and therefore were subjected to non-parametric tests? This is not clear from the descriptions provided.
Discussion section: In restating the salient results from this study, the authors should highlight new findings not been observed previously in studies of ethanol and dietary iron.
The authors should also provide a caveat that most of the experimental measures examined in this study were morphologic or hematological in nature and that biochemical and/or physiological measures of organ system function may be affected to a greater degree by ethanol exposure or by iron supplement intervention.
Reviewer 2 Report
In this article Helfrich et al. demonstrated that gestational administration of the clinically relevant iron supplement mitigates alcohol’s adverse impacts upon the fetus by using Long-Evans rats. They found that alcohol consumption caused fetal anemia, decreased fetal body and brain weight, increased hepatic iron content, and elevated hepatic malondialdehyde. Supplemental iron normalized this brain weight reduction in alcohol-exposed males but not female littermates. Iron also reversed the alcohol-induced fetal anemia and normalized both red blood cell numbers and hematocrit suggesting that gestational iron supplementation could be a clinically feasible approach to improve prenatal iron status and fetal outcomes in alcohol-exposed pregnancies.
The manuscript is clear and generally well written.
Author Response
Reviewer 2 had no concerns regarding the manuscript.
Reviewer 3 Report
The most important finding from this study iss that gestational iron supplementation selectively reversed the negative effects of PAE upon fetal anemia and male brain weight. These effects paralleled but were not as strong or extensive as the effects of dietary iron fortification (IF) in PAE, although the chemical form (ferrous sulfate) was the same. It is a good paper.
Discussion appears to be best part of the entire paper. However, it may be written with a better flow, avoid repeating concepts that have been mentioned in introduction or elsewhere.
"We selected this iron dose because the equivalent iron dose 116
in humans (300-600 mg/d) is at the upper end of the therapeutic dose of iron as a treatment 117 for ID." Could you insert any reference?
The tables and figures could be improved. It is difficult understand the meaning of statistical signals.
Author Response
Reviewer 3
Discussion appears to be best part of the entire paper. However, it may be written with a better flow, avoid repeating concepts that have been mentioned in introduction or elsewhere.
Response: The reviewer suggested that the Discussion could have been written with “a better flow” but did not provide specifics. We have trimmed elements of the Discussion to reduce wordiness.
"We selected this iron dose because the equivalent iron dose 116
in humans (300-600 mg/d) is at the upper end of the therapeutic dose of iron as a treatment 117 for ID." Could you insert any reference?
Response: The reference for the iron dose selection was already present as #49, the US DRIs for iron. We edited this sentence for clarity.
The tables and figures could be improved. It is difficult understand the meaning of statistical signals.
Response: Meaning of statistical signals. We edited the figure and table legends for clarity.
Round 2
Reviewer 1 Report
The authors have done a good job of responding to the reviewers' comments on the original manuscript. This reviewer appreciates the efforts to enhance the clarity of the data presentation as well as some statements within the text of the revised manuscript.